# Mucoadhesive In Situ Rectal Gel Loaded with Rifampicin: Strategy to Improve Bioavailability and Alleviate Liver Toxicity

**DOI:** 10.3390/pharmaceutics13030336

**Published:** 2021-03-05

**Authors:** Fakhria Al-Joufi, Mohammed Elmowafy, Nabil K. Alruwaili, Khalid S. Alharbi, Khaled Shalaby, Shakir D. Alsharari, Hazim M. Ali

**Affiliations:** 1Department of Pharmacology, College of Pharmacy, Jouf University, Sakaka P.O. Box 2014, Saudi Arabia; faaljoufi@ju.edu.sa (F.A.-J.); kssalharbi@ju.edu.sa (K.S.A.); 2Department of Pharmaceutics, College of Pharmacy, Jouf University, Sakaka P.O. Box 2014, Saudi Arabia; Nkalruwaili@ju.edu.sa (N.K.A.); khshalabi@ju.edu.sa (K.S.); 3Department of Pharmaceutics and Industrial Pharmacy, Faculty of Pharmacy (Boys), Al-Azhar University, Nasr City, Cairo P.O. Box 11651, Egypt; 4Department of Pharmacology and Toxicology, College of Pharmacy, King Saud University, Riyadh P.O. Box 11451, Saudi Arabia; sdalsharari@ksu.edu.sa; 5Department of Chemistry, College of Science, Jouf University, Sakaka P.O. Box 2014, Saudi Arabia; hmali@ju.edu.sa; 6Forensic Chemistry Department, Forensic Medicine Authority, Cairo P.O. Box 11614, Egypt

**Keywords:** rifampicin, mucoadhesive, rectal in situ gelling, hepatotoxicity

## Abstract

Although it is a front-line in tuberculosis treatment, rifampicin (RF) exhibits poor oral bioavailability and hepatotoxicity. Rectal mucoadhesive and in situ rectal gels were developed to overcome drug drawbacks. A RF/polyethylene glycol 6000 co-precipitate was first prepared in different ratios. Based on the drug solubility, the selected ratio was investigated for drug/polymer interaction and then incorporated into in situ rectal gels using Pluronic F127 (15%) and Pluronic F68 (10%) as a gel base and mucoadhesive polymers (HPMC, sodium alginate and chitosan). The formulations were assessed for gelation temperature and gel strength. The selected formulation was investigated for in vivo assessments. The results showed that a 1:1 drug/polymer ratio exhibited satisfying solubility with the recorded drug/polymer interaction. Depending on their concentrations, adding mucoadhesive polymers shifted the gelation temperature to lower temperatures and improved the gel strength. The selected formulation (F4) did not exhibit any anal leakage or marked rectal irritation. Using a validated chromatographic analytical method, F4 exhibited higher drug absorption with a 3.38-fold and 1.74-fold higher bioavailability when compared to oral drug suspension and solid suppositories, respectively. Toxicity studies showed unnoticeable hepatic injury in terms of biochemical, histopathological and immunohistochemical examinations. Together, F4 showed a potential of enhanced performance and also offered lower hepatic toxicity, thus offering an encouraging therapeutic alternative.

## 1. Introduction

Tuberculosis (TB) is an infectious illness produced by Mycobacterium tuberculosis and its extermination is still one of the chief challenges in current public health [1]. According to WHO report in 2019, TB endures as the top infectious killer worldwide, with 10 million people falling ill with TB and 1.5 million deaths in 2018 [2]. Especially if the patient is suffering from acquired immunodeficiency syndrome (AIDS), the mortality rate increases. Although the active drugs are available for treatment, the full treatment strategy is still facing hurdles involving long-term treatment, reduced patient compliance, drug-associated toxicity and multidrug resistance [3].

Rifampicin (RF) is a semisynthetic macrocyclic antibiotic and one of first-line antituberculosis drugs. It acts by inhibition of bacterial RNA synthesis. However, RF suffers from many drawbacks, such as a poor and unreliable pharmacokinetic profile, the short plasma half-life of rifampicin (2 h) after oral administration [4], severe hepatotoxicity, acid degradation and patient incompliance in long-term drug therapy, due to the necessity to administer a daily dose or doses several times per week which leads to treatment failure and the occurrence of drug resistance [5]. Thus, several studies have investigated the absorption efficiency in an effective and safe way. Bachhava et al. developed RF-loaded lipomers to improve Peyer’s patch uptake, and hence, oral delivery [6]. Singh et al. developed RF-loaded phospholipid liposphere carriers for pulmonary application in order to improve pulmonary drug delivery via biocompatible (polymer-free) carriers [7]. However, RF-induced hepatotoxicity has been attenuated by complexation with phospholipids [8]. The authors attributed the mitigation of hepatotoxicity in the murine model to the antioxidant and hepatoprotective effects of phospholipids.

As an alternative route to oral delivery, rectal absorption efficiency is recommended to overcome the problems associated with bioavailability and therapeutic efficacy [9]. A conventional solid suppository melts or softens in the rectal area at normal body temperature. It has several disadvantages that include patient discomfort, low patient compliance and anal leakage. In addition, it can reach upper hemorrhoidal vein and suffer the first-pass effect [10]. Therefore, the ideal suppository should not induce pain throughout application or leak from the anus. It should also remain at the administration site without further upward travelling to bypass the first-pass effect [11]. Therefore, in situ rectal gel is considered to be a better substitute for a conventional solid suppository as the latter is converted into gel form at physiological body temperature. Pluronics (poloxamers): a copolymer of poly(oxyethylene)–poly(oxypropylene)–poly(oxyethylene) that offer a distinct base in terms of good tolerability, inducing minimal irritation and sensitivity on the skin, and thus far have been found to be valuable in rectal application [12]. The most key property of Pluronics is thermosensitivity, which means that Pluronic-based formulations can be introduced to the patient in the form of a solution and convert into gel upon rectal administration, allowing the patient better convenience and minimal drug leakage. Additionally, Pluronics were reported to improve bioavailability of ibuprofen [13], nimisulide [14], flurbiprofen [15] and ondansetron [16] after rectal administration. 

As this type of suppository is mainly Pluronic based, adding mucoadhesive polymer is easy and can improve attachment to rectal mucosa. In that sense, we can improve drug bioavailability, avoid stomach passing, and hence, acid degradation. It also can bypass liver metabolism and alleviate the induced toxicity.

Keeping in mind the aforementioned issues, the objective of this work was to fabricate RF loaded in situ rectal gels in order to minimize acid degradation of the drug following oral administration, improve RF bioavailability and alleviate RF-induced hepatotoxicity. RF solubility was first enhanced by the co-precipitate formulation and then the co-precipitate was incorporated into Pluronic-based in situ rectal gels along with the addition of different mucoadhesive polymers to maximize the effect. The selected formulation was exposed to bioavailability and toxicity studies and compared with oral suspension and conventional solid suppositories.

## 2. Materials and Methods

### 2.1. Materials

Rifampicin (RF) was obtained from Central Drug House (CDH, purity 97%, Vardaan House, New Delhi, India). Pluronic F127, Pluronic F68 and sodium alginate (SA; molecular weight: 80,000–120,000 Da with a mannuronic/guluronic ratio of about 1.56; viscosity: 3.500 cP) were purchased from Sigma Aldrich (Shanghai, China). Hydroxypropyl methyl cellulose (HPMC) was purchased from Loba Chemie (Mumbai, India). Chitosan (molecular weight: 100,000–300,000 Da; degree of deacetylation: 85%; viscosity: 200–800 cP) was purchased from ACROS organics (Thermo Fischer Scientific, Bedford, MA., USA). Witepsol H15 was kindly gifted by DELTA Pharmaceutical Company (Cairo, Egypt). Polyethylene glycol 6000 (PEG 6000) was obtained from Techno Pharmchem (New Delhi, India). The other chemicals and solvents in the study were of analytical grade.

### 2.2. Development of RF Co-Precipitate

The mixtures of RF and PEG 6000 (1:0.25, 1:0.5, 1:1 and 1:2 w/w) were dissolved in chloroform using a magnetic stirrer at 300 rpm for 15 min at ambient temperature (Wisestir, MH-20D digital magnetic stirrer, Seoul, Korea). Organic solvent was eliminated by vaporization at 45 °C for 2 h using a rotary evaporator (Buchi, Flawil, Switzerland). The residues were further dried in air overnight and pulverized. Physical mixtures were developed by blending appropriate amounts of RF and PEG 6000 using a mortar and pestle.

### 2.3. Determination of RF Co-Precipitate Aqueous Solubility

To establish the most favorable ratio of RF to PEG 6000, a solubility check was carried out by adding different prepared co-precipitate formulations into 10 mL of double distilled water. Samples were put in a thermostatic shaker (Julabo SW22 GmbH, Seelbach, Germany) at ambient temperature adjusted at 50 rpm for 48 h. Samples were then filtered and the concentrations of RF were determined using by UV spectroscopy at λ_max_ 475 nm. The solubility of the pure RF and the physical mixture were also determined.

### 2.4. RF/PEG 6000 Interaction

#### 2.4.1. Thermal Analysis Using Differential Scanning Calorimeter (DSC)

RF, PEG 6000, the co-precipitate and the RF/PEG 6000 physical mixture were tested for thermal analysis by a DSC (DSC3, Mettler Toledo, Greifensee, Switzerland) equipped with STAR^e^ 15.00 software in order to check the RF/polymer interaction in terms of the sample transition temperature. Six milligrams of each sample were accurately weight and put in crucible aluminum pans which were directly closed. The samples were analyzed between 25 °C and 200 °C using a blank aluminum pan as the reference. Thermograms were run at a heating rate of 20 °C/min under flow of nitrogen gas. 

#### 2.4.2. Fourier Transform Infrared Spectroscopy (FTIR)

We investigated the interaction between RF and PEG using the FTIR technique (Lambda Scientific, Edwardstown, Australia). The samples were separately mixed with anhydrous potassium bromide and pressed to form thin films. The films were hung at an apparatus holder and analyzed between 4400 cm^−1^ and 350 cm^−1^ in the transmission mode.

### 2.5. Preparation of in Situ Gels

The cold method [17] was utilized to incorporate the RF co-precipitate into the thermosensitive gel base. RF in situ rectal gels were fabricated using Pluronic F127 (15%) and Pluronic F68 (10%) as a gel base. The RF co-precipitate was first dissolved in a calculated amount of double distilled water using a magnetic stirrer followed by the slow addition of various percentages of mucoadhesive polymers (HPMC, sodium alginate and chitosan). HPMC and sodium alginate were dissolved in distilled water, while chitosan was dissolved in a 0.25% acetic acid solution. With continuous stirring, Pluronic F127 and Pluronic F68 were slowly added at room temperature. The prepared gels were kept overnight in the refrigerator to obtain a clear solution. The formulations were designed to obtain 50 mg/mL of RF. The detailed composition of various batches is illustrated in Table 1.

### 2.6. Gelation Temperature

The gelation temperature of the gel base, the RF co-precipitate in the gel base and the RF co-precipitate in the mucoadhesive gel base were determined by the same method described by Yun and co-workers [18], with minor modifications. Concisely, 15 g of each formulation was positioned in a 20 mL transparent glass vessel with a magnetic bar. The vials were put in a thermostatted water bath. The temperature was gradually elevated by 1 °C/min, starting from 22 °C, with constant stirring (50 rpm). Gelation temperature was considered when the magnetic bar stopped moving [19]. The experiment was performed in triplicate.

### 2.7. Gel Strength Determination

The gel strength of the investigated formulations was investigated by the procedure reported by Barakat in 2009 [20] with a slight modification. Briefly, 50 g of in situ gelling formulations were placed in a 100 mL graduated cylinder, which was put in a water bath. The temperature was equilibrated at the corresponding gelation temperature of each formulation. A disk of 1.5 cm diameter and weighing 35 g was positioned on the top of the formulations in the cylinder. The time (s) passed to drop the disk by 5 cm down was recorded and considered as an arbitrary index of gel strength. If the time taken for the disk to fall by 5 cm was more than 5 min, different weights were used and the gel strength was defined by the least amount of weight that pushed the device 5 cm down through the gel [21].

### 2.8. Animals

For all in vivo studies, male albino rabbits weighing 2.5–3 kg were used. All animals were habituated at ambient temperature (25 ± 2 °C) with free access to diet and drinking water. The work was performed in accordance with ethics and guidelines of Local Committee for Research Bioethics, Jouf University (Ethical code; LCBE03/04/41, 23 February 2020).

### 2.9. In Vivo Rectal Retention

To examine rectal retention, four albino rabbits were rectally administered F4 4 cm above the anus via a plastic syringe [21].

### 2.10. Investigation of Rectal Irritation

The safety of the rectal administration of the selected formulation was assured by the examination of rectal specimens and detection of histopathological changes (rectal irritation). Eight male albino rabbits were allocated into two groups (four animals each). They were fasted for 12 h before the experiment with free access to water. The first group was considered as the untreated control group and given saline, while the second group received F4 rectally 4 cm above the anus at a dose of 35 mg/kg. After 24 h, the animals were killed and the rectal tissues were removed, rinsed with saline solution and placed in 10% neutral phosphate-buffered formaldehyde. The fixed samples were dehydrated through different alcohol grades and fixed in paraffin. Serial sections of 4 µm thickness were stained with hematoxylin and eosin (H&E). The sections were then inspected by a light microscope for histological changes.

### 2.11. Pharmacokinetics Study

A pharmacokinetics experiment was performed to investigate the potential of in situ rectal gel to improve RF bioavailability when compared to conventional solid suppositories and oral suspensions. Animals were distributed into three groups (six animals for each group):

Group 1: administered the RF suspension (made by dispersing RF in 0.5% carboxymethylcellulose sodium) as a single oral dose. 

Group 2: administered a single rectal dose of solid suppository containing Witepsol H15 as the suppository base. Solid suppositories were fabricated by the melting method, in which Witepsol H15 was melted followed by addition of RF at 37 °C with continuous agitation until a homogenous dispersion was obtained. The dispersion was allowed to cool at room temperature, and then stored at 4 °C. 

Group 3: administered a single rectal dose of F4.

All groups were given an administered dose equivalent to 35 mg of RF per kg. Blood samples (1 mL) were withdrawn from the sinus orbital at 0.5, 1, 2, 3, 4, 6, 8, 12 and 24 h after administration. After that, blood samples were withdrawn into heparinized tubes followed by centrifugation at 5000× *g* for 15 min for plasma separation. Plasma samples were kept at −40 °C until RF quantification. 

Different pharmacokinetic parameters were obtained by Kinetica™ v.4 software. After the plasma level–time curve construction, the time passed to reach the highest plasma concentration (T_max_) and maximum plasma concentration (C_max_) were directly calculated. The linear trapezoidal method was utilized to determine the area under the curve (AUC). The relative bioavailability of solid suppositories and F4 were calculated using the following equation:$$\mathrm{Relative}\;\mathrm{bioavailabily}\;\left(\mathrm{RB}\right)=\;\frac{\mathrm{AUC}\;\mathrm{of}\;\mathrm{solid}\;\mathrm{suppositories}\;\mathrm{or}\;F4\;\times \;\mathrm{dose}\;\mathrm{of}\;\mathrm{oral}\;\mathrm{suspension}}{\mathrm{AUC}\;\mathrm{of}\;\mathrm{oral}\;\mathrm{suspension}\;\times \;\mathrm{dose}\;\mathrm{of}\;\mathrm{solid}\;\mathrm{suppositories}\;\mathrm{or}\;F4}\;\;\;$$

### 2.12. Chromatographic Conditions

The chromatographic quantification of RF was carried out using an ultra-high-performance liquid chromatography (UHPLC) system (Thermo Scientific Dionex UltiMate 3000UHPLC+, Thermo Fischer Scientific, Bedford, MA, USA) and focused standard systems operated with a pump, degasser, analytical split-loop thermostatted well-plate autosampler, thermostatted column compartment, solvent rack without a degasser and diode array detector. The chromatographic data were acquired and processed using Chromeleon™ 7.2 software (Thermo Fischer Scientific). The separation was performed at Thermo Scientific ACCLAIM™ 120 C-18 column (150 mm × 4.6 mm i.d and 5 μm of particles size). The mobile phase was composed of a mixture of acetonitrile (solvent A) and 0.01% formic acid in water (solvent B) (75: 25 *v*/*v*) using isocratic elution at a flow rate of 0.8 mL/min. The column temperature was set at 25 °C, the injection volume was 10 µL and the total time of analysis was set at 6 min. Validation of the method was assessed in terms of linearity, accuracy and precision, see Appendix A.

### 2.13. Sample Preparation

For the extraction of RF from plasma and protein precipitation, an aliquot of 200 µL of plasma-containing RF was transferred into an Eppendorf tube, 300 µL of acetonitrile and 500 µL of methanol were added, the mixture was shaken well for 30 s on a Vortex mixer (DRAGONLAB, Beijing, China), and then subjected to centrifugation for 5 min at 6000 rpm. Afterwards, 10 µL of the supernatant was injected into the UHPLC system.

### 2.14. Toxicity Studies

In order to investigate safety of F4 when compared to conventional solid suppositories and oral suspensions, toxicity studies were done. Rabbits were allocated into four groups. Group I received a saline solution, while groups II, III and IV received an oral RF suspension, a rectal conventional solid suppository and rectal F4 once daily, respectively. After 2 months, blood samples were taken for the evaluation of liver function by determining the liver enzymes (serum amino transferases; alanine amino transferase (ALT) and aspartate amino transferase (AST)), albumin and total bilirubin. Additionally, the animals were sacrificed, and the liver tissues’ specimens were gathered for the determination of histological alterations and immunohistochemistry. For histological changes, liver specimens were fixed and stained with H&E using the same procedure mentioned above in the rectal irritation section (Section 2.10). The immunohistochemical (IHC) investigation was done by determining the tumor necrosis factor α (TNF-α) immunoexpression in paraffin sections by adding 10% goat serum to block unspecific binding, followed by adding anti-TNF-α antibodies [22]. The sections were then counterstained with hematoxylin and inspected under a light microscope. 

### 2.15. Statistical Analysis

All results were analyzed by one-way ANOVA and the means were compared by Tukey’s multiple comparison testing using GraphPad Prism v.5. Software (GraphPad Software, San Diego, USA). A difference at *p* < 0.05 was considered to be significant.

## 3. Results and Discussion

### 3.1. RF Co-Precipitate Aqueous Solubility

As RF is slightly soluble in water, direct loading into in situ rectal gels results in the formation of an unstable suspension that is difficult to redisperse [14]. Therefore, enhancing RF solubility is essential before incorporating it into in situ rectal gel formulations. One of the most applicable methods is solid dispersion [23] by drug carrier co-precipitation. However, PEG is frequently used carrier for this purpose, as it also improves drug stability, and it is also used via molecular interaction [24]. Thus, a RF-PEG co-precipitate was prepared in order to increase RF solubility. Different ratios of RF to PEG 6000 (1:0.25, 1:0.5, 1:1 and 1:2 *w*/*w*) were used to obtain the optimal ratio that enhanced RF solubility to a level that enabled us to incorporate the RF therapeutic dose into gel without separation. Figure 1 shows the solubility of different co-precipitate batches. It is clear that RF aqueous solubility was about 2.6 mg/mL, which was similar to the RF-PEG physical mixture, confirming the absence of a reaction between RF and PEG. It was also obvious that the co-precipitation of RF with PEG particularly enhanced RF aqueous solubility. RF solubility was dependent on PEG concentration; the higher the PEG concentration, the higher the RF solubility. Duplication of the solubility was achieved by a 1:1 RF to PEG ratio in the co-precipitate. As this solubility is suitable for incorporation into the in situ rectal gel base and in vivo dose, the 1:1 ratio of RF:PEG 6000 was designated for further characterization.

### 3.2. Thermal Analysis

To examine the level of crystallinity of the co-precipitate and the influence of the RF addition, DSC analyses were performed (Figure 2). RF exhibited a melting endotherm in the range of 180 °C to 200 °C (186.3 °C), directly followed by an exothermal recrystallization at 256.2 °C. Agrawal et al. (2004) reported that RF in form II exhibited an endothermic peak in the range of 180 °C to 200 °C, which was followed by an exothermal recrystallization in the range of 200 °C to 225 °C (transformation into form I). After that, RF decomposed in the range of 230 °C to 280 °C [25]. In our study, PEG 6000 showed sharp main peaks at 65.5 °C. The exotherm detected for RF material in the physical mixture was confirmed near its known melting point, although it was faintly depressed and moved to a lower temperature (251.4 °C). The depression of this peak and cleavage of the PEG 6000 peak may be ascribed to the dilution of RF in the physical mixture and also the high solubilization capacity of PEG 6000 toward RF. In the co-precipitate powder, the endothermic peak of RF was completely undetectable, rather, it exhibited two peaks; endothermic at 63.5 °C and depressed exothermic at 245 °C. Shifting the recrystallization exothermic peak into a lower temperature confirmed the more stable amorphous state of these samples. It was reported that when the crystallinity of the drug is decreased, its release/solubility could be enhanced correspondingly, owing to the decreased energy required for breaking the crystal lattice arrangement during the dissolution process [26].

### 3.3. FTIR Analysis

The interaction of RF and PEG 6000 was further investigated by the FTIR tool. FTIR analysis detects the specific absorption of IR beams at definite chemical bonds. The interaction is confirmed by a change in either vibration intensity or vibration frequency [27]. The FTIR spectra of pure RF, PEG 6000, the physical mixture and the co-precipitate are presented in Figure 3. The RF spectrum showed distinctive peaks at 1728 cm^−1^ (revealing carbonyl groups), 1650 cm^−1^ (revealing of -C=N- asymmetric stretching group) and 1566 cm^−1^ (revealing of the amide II group). The electrostatic interaction between the PEG 6000 and RF was confirmed by the depression of some characteristic peaks (at 1728 cm^−1^ and 1650 cm^−1^) and the disappearance of the characteristic peak (at 1566 cm^−1^), as illustrated by dashed lines in solid dispersion, indicating the complexation of RF with PEG 6000.

### 3.4. Gelation Temperature

The gelation temperature is recognized as the temperature at which the liquid state transforms into gel form. For in situ rectal gels, it is preferred to have a gelation temperature ranging from 30 to 36 °C. However, gelation temperature is considered to be one of the chief characteristics of in situ rectal gels and its determination is considered to be a cornerstone in its effectiveness. Gelation below body temperature and near room temperature will cause production, handling and administration difficulties, while gelation above body temperature will deprive the suppositories from their constitutional role and superior in vivo efficacy. In addition, they will remain in a liquid state at body temperature, leading to liquid escaping from the anus. Table 2 depicts the gelation temperatures of different batches. Depending on this hypothesis and our trials, we selected Pluronic F127 and Pluronic F68 in a ratio 15:10 as a base mixture because it showed a gelation temperature of 36.1 °C. Pluronic molecules have the ability to arrange in a zigzag-like configuration, and become more closely packed with temperature increases, resulting in a more viscous system [28]. Thermoreversible behavior is ascribed to interactions between different segments, such as temperature elevation, dehydrated polypropylene oxide blocks and micellization. This micellization is followed by gelation because of the ordered packing of the micelles. On the other hand, the addition of a RF co-precipitate markedly increased the gelation temperature of the base (43.8 °C). This increase in gelation temperature might be attributed to presence of PEG 6000 in the co-precipitate added to the base, which was demonstrated to interfere the process of micellar association of Pluronic chains [29]. On the other hand, adding bioadhesive polymers is beneficial, as they can reinforce the gel viscosity and bioadhesive force, and shift the gelation temperature into lower degrees. The mucoadhesive polymers used were hydroxypropyl methyl cellulose (HPMC; as a neutral polymer), sodium alginate (as an anionic polymer) and chitosan (as a cationic polymer). The influence of bioadhesive polymers on the gelation temperature relied upon the nature of the bioadhesive additives (polymers) and their concentrations in the formulations. Bioadhesive polymers had a concentration dependent T_sol–gel_ decreasing effect (Table 2). The most marked lowering effect was recorded upon the addition of sodium alginate at a 1.2% concentration. It lowered the T_sol–gel_ by 9.6 °C when compared to the mucoadhesive polymer-free formulation (F1). In addition, 1% HPMC decreased the gelation temperature by 6 °C. The gelation temperature-decreasing effect of bioadhesive polymers might be elucidated by the capability of the polymers to connect to polyethylene oxide (PEO) chains present in the Pluronic molecules, enhancing the dehydration effect and producing an increment in the entanglement of neighboring molecules with more marked intermolecular hydrogen bonding [30]. Mayol and coworkers studied the effect of the addition of bioadhesive polymers (hyaluronic acid), utilizing the tools of differential scanning calorimetry (DSC) and thermogravimetric analysis (TGA). They found that adding such polymers into Pluronic gels hampered the interactions between the water and Pluronic molecules, where the water was strongly bound to the gel network. By raising the temperature, Pluronic micelle assembly and packing were enhanced, resulting in the improvement of the gelation process and lowering of the T_sol–gel_ [31].

### 3.5. Gel Strength Determination

Another important parameter is gel strength, which determines the easiness of insertion of the suppositories and allows for minimum or no anal escape. It was previously demonstrated [21] that the ideal in situ rectal gels should have appropriate gel strength (10–50 s). Similar to transition temperature, the gel strength is particularly influenced by the concentration, nature and composition of bioadhesive polymers (Table 2). All bioadhesive polymers increased the gel strength in a concentration-dependent manner. This increment was reported to be due to the binding of the mucoadhesive polymer by hydrogen bonding to the cross-linked Pluronic gel [13]. This result was in good accordance with Yong et al. [32], who claimed that sodium alginate (in a 0.2–1.0%. concentration) had a pronounced effect on the gel strength of acetaminophen in situ rectal gels. HPMC showed the most pronounced increase in the gel strength (29.5–47.8 s), upon increasing the concentration from 0.5 to 1.5%.

Based on the above-mentioned results, we selected F4 (consisting of 10% of Pluronic F68, 15% of Pluronic F127, 0.1% of the co-precipitate and 1.2% of sodium alginate) for the in vivo experiment, as it showed encouraging results of both gelation temperature and gel strength.

### 3.6. Rectal Retention

The investigation of rectal retention can indirectly reflect the mucoadhesion characteristics of the developed formulae. Anal leakage commonly takes place after the rectal administration of conventional solid suppositories. F4 did not exhibit any anal leakage, which reflected its mucoadhesive force. Sodium alginate was considered as a predominant factor in the rectal retention of F4. Sodium alginate is an anionic water-soluble mucoadhesive polymer. Its structure is constituted from linear polysaccharide of 1–4 linked α-L-guluronic acid and β-D-mannuronic acid, so it can form strong hydrogen bonds with mucosal mucin glycoproteins through carboxyl–hydroxyl interactions [33].

### 3.7. Rectal Irritation

To investigate the safety of F4, we assessed rectal mucosal damage/irritation by the histological examination of rectal tissues after the administration of F4. Figure 4 depicts the morphology of the rectal mucosa of the normal control rabbits (Figure 4A) and after the rectal administration of F4 (Figure 4B). The normal control showed normal mucosal structure. The F4-treated group showed no evidence of remarkable epithelial necrosis or hemorrhage. It showed only a few eosinophils infiltrating the rectal mucosa. Furthermore, the morphology and structure of the rectal mucosa or colonic gland layer looked like the control group, which confirmed the safety of F4. These results are in good accordance with Ryu et al., who confirmed the safety of Pluronics and sodium alginate [34].

### 3.8. Pharmacokinetic Studies

Pharmacokinetic studies were performed after the administration of RF by both oral (RF suspension) and rectal (solid suppository and in situ rectal gels (F4)) routes. The plasma level-time curves of different groups are shown in Figure 5, and the pharmacokinetic parameters of these groups are outlined in Table 3. It was noticed that F4 reached its maximum plasma concentration (*T*max) more rapidly (after 2 h) than the oral suspension and solid suppository (3 h). This reflected the changeability in absorption among different investigated formulations. Particularly for poorly water-soluble drugs, the incorporation into fatty-based suppositories could slow down their release into rectal fluid, because such drugs prefer a base matrix over the aqueous rectal fluid. Moreover, and regardless of peak plasma concentration values, the plasma concentrations of RF in the case of F4 were higher than the corresponding time points of that of the oral suspension and conventional solid suppository. In particular, in F4, from 0.5 h to 3 h, the plasma concentrations of RF (12–31 μg/mL) were significantly higher than those of the solid suppository (6–25 μg/mL) and oral suspension (4–17 μg/mL). Higher plasma concentrations in the case of F4 might be attributed to the dispersibility (fluidity) and bioadhesive force variation [14]. In the case of the in situ gel, the dispersion was more pronounced in the rectal fluids as RF was introduced in a soluble co-precipitate form. Bioadhesion and prolonged retention were also higher due to the effect of sodium alginate, which improved attachment to the rectal mucosa [21]. In the case of the conventional solid suppository, RF was insoluble in aqueous fluid and preferred to exist in the lipophilic fatty media of the conventional solid suppository than to escape into aqueous rectal media. The formulation was also deprived from the bioadhesive force. In the case of the RF oral suspension, pure RF was poorly soluble and expected to undergo acid sensitivity [35]. The absorption half-life (T_1/2ab_) of F4 (4.95 ± 0.35 h) was higher than other formulations (0.15 ± 0.07 h and 1.07 ± 0.93 h for conventional solid suppository and oral suspension, respectively), which might be due to the extended release of RF, owing to the bioadhesive gel matrix of Pluronics and sodium alginate created at body temperature. A longer elimination half-life (T_1/2el_) was observed for F4 when compared to the conventional solid suppository and oral suspension, which could assist in decreasing the frequency of the dosing. In turn, this also reflected a higher residence time of F4 (7.86 ± 1.1 h) when compared to the conventional solid suppository (6.54 ± 0.27 h) and oral suspension (5.38 ± 0.84 h). F4 provided a significantly (*p* < 0.05) higher AUC_0–24_ for F4 (293.76 ± 21.7 µg·h/mL) than the conventional solid suppository (151.70 ± 12.4 µg·h/mL) and oral suspension (86.76 ± 6.8 µg·h/mL). On the basis of these results, we summarized the probable mechanism of the enhanced bioavailability of RF by employing the mucoadhesive in situ gelling rectal suppository formulation when compared to the conventional solid suppository and oral suspension. First, the rectal route can generally protect the drug from the acid degradation followed by oral administration. Second, the prefabrication of the co-precipitate and the base (Pluronics [12]) enhanced RF solubility, which was unlikely to take place in the limited volume of the rectum (as in the case of conventional solid suppositories) without improving the dissolution through such a technique. Third, the mucoadhesion effect of the added polymer and the base (Pluronics) prevented travelling into the upper hemorrhoidal vein and improved lower rectum retention. Absorption via the lower hemorrhoidal vein could often improve drug bioavailability. In addition, the base constituents (Pluronic F127) [36] can promote the stabilization of susceptible drugs when incorporated in their micelles [37].

### 3.9. Toxicity

The toxicity of RF was assessed for the oral RF suspension, rectal solid suppository and rectal F4 via biochemical, histopathological and immunohistochemical (IHC) investigations after 2 months of daily dosing.

#### 3.9.1. Biochemical Changes

Table 4 illustrates the biochemical changes in various groups including ALT, AST, albumin and total bilirubin. ALT and AST are enzymes that are used as liver function markers [38]. They increase in liver injuries as they leak from hepatocytes, leading to high plasma levels. Albumin is specifically synthesized in the liver [39], whereas the elevation of total bilirubin indicates the depth of jaundice. Rabbits that received the oral RF suspension showed significant (*p* < 0.05) changes in ALT (80.3 ± 0.16 U/L), AST (37.97 ± 5.9 U/L), albumin (3.41 ± 0.18 mg/dL) and total bilirubin (0.61 ± 0.06 mg/dL) when compared to the control group. The mechanism by which RF can induce liver toxicity is not fully established, but it can be partially explained by two mechanisms: the first is that the liver is the major site for metabolizing antitubercular drugs [40] and for the production of RF metabolites, which may have direct toxicity or induce immunological liver injury [41]; and the second is hepatic enzyme induction, which enhances the metabolism of other co-administered drugs. However, the toxicity is mainly accompanied by a significant increase in liver enzymes [42] and decrease in albumin [43] and cholestasis [44]. Furthermore, RF can inhibit bilirubin secretion of bile duct and then elevate bilirubin levels in serum [45]. A significant decrease in ALT and total bilirubin was observed in the group administered F4 when compared with the oral suspension group (*p* < 0.05). Liver toxicity is supposed to be due to the formation of an idiosyncratic reaction to RF metabolites, followed by hepatic metabolism in the case of the RF suspension. Even solid suppositories have the same route of administration of F4, and the formulation solid suppositories did not exhibit the significant improvement. However, an additional evaluation of formulation-induced toxicity was assessed by histology and immunohistochemistry.

#### 3.9.2. Histopathological Examination

The examination of liver sections of the control rabbits exhibited a normal liver histological picture with distinct hepatocytes (Figure 6A,B). On the other hand, liver sections of the group administered the oral RF suspension showed severe and diffuse hydropic degeneration in hepatocytes with congested central veins (arrow in Figure 6C), pyknotic nuclei of swollen hepatocytes, few perivascular mononuclear cell infiltration and focal replacement of hepatic parenchyma with mononuclear cell infiltration (arrow in Figure 6D). RF can initiate liver injuries because of oxidative stress in the mitochondria and hepatic lipid accumulation [46]. In another study, RF-induced liver toxicity was attributed to reactive metabolite formation [45]. The examination of liver sections of the rabbits that received solid suppositories showed moderate improvement of the hepatic histological picture, in which slight congestion in the central veins (arrow in Figure 7A) and moderate hydropic degeneration in hepatocytes in the centrilobular zone (Figure 7A,B) with little mononuclear cell infiltration in the portal area (arrow in Figure 7B) were obvious. Meanwhile, the in situ gel (F4) administered group revealed the liver histology was kept near normal structure, in which mild hepatocyte swelling in the centrilobular zone with narrowing of the hepatic sinusoids and diffuse Kupffer cell proliferation (Figure 7C,D) were observed. The superiority of the mucoadhesive in situ rectal gel (F4) was evidenced in liver bypassing, as mucoadhesion could restrict the formulation from travelling into upper parts of the rectum. This could partly explain the difference between the solid suppositories and F4. 

#### 3.9.3. Immunohistochemistry

TNF-α is a pleiotropic cytokine produced by activated immune cells and recognized to offer a crucial part in the pathogenesis of liver injury associated with drug use. It applies inflammatory, cytotoxic, angiogenic and growth modulatory effects on different target cells [47]. It also motivates the production of nitric oxide (NO^•^) contributing to nitrosative stress. NO^•^ could also react with superoxide (O_2_^•^^−^), resulting in the formation of peroxynitrite (ONOO^−^), both of which initiate cellular hazards. As TNF-α is significantly elevated in livers treated with isoniazid and rifampicin [48,49], we estimated its production via immunohistochemical investigation. Inspection of the control group revealed that TNF-α was nearly at a zero scoring (Figure 8A). The sections of the animals that received RF suspension exhibited an intensive positive reaction (brown color) in hepatocytes, endothelial cells and lymphocytic aggregation (Figure 8B). In sections of the animals that received solid suppositories, a slightly increased positive reaction in hepatocytes was observed (Figure 8C). Meanwhile, the sections of the group that received the mucoadhesive in situ rectal gels formulation (F4) showed a very mild positive reaction in hepatocytes (Figure 8D). The morphometric analysis (Figure 9) showed that the area percentage of the TNF-α reaction was significantly (*p* < 0.05) increased in the hepatic sections in the group that was administered the RF suspension. This TNF-α overexpression reflected a condition of tissue inflammation. Regarding the group receiving F4, the area percentages were significantly (*p* < 0.05) decreased, indicating the safety of the formulation. Collectively, the mucoadhesive in situ rectal gel formulation (F4) showed a promise of better performance and also proposed lower liver toxicity, thereby presenting an encouraging therapeutic alternative.

## 4. Conclusions

Our study established a formulation of a RF-loaded mucoadhesive in situ rectal gel as a rational approach for improved RF delivery and liver toxicity attenuation. RF solubility was first enhanced via the formation of a co-precipitate, and then incorporated into thermosensitive mucoadhesive in situ rectal gels. The selected formulation (consisting of 10% of Pluronic F68, 15% of Pluronic F127, 0.1% of the co-precipitate and 1.2% of sodium alginate) showed enhancement in both the drug absorption and alleviation of drug-induced liver toxicity. Such a formulation could offer a therapeutic alternative for improved therapy of pulmonary tuberculosis.

## Figures and Tables

**Figure 1 pharmaceutics-13-00336-f001:**
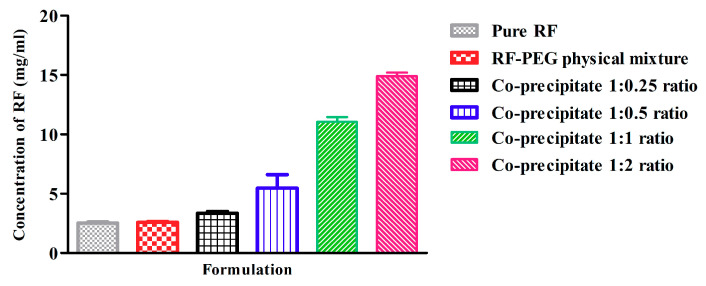
Solubility of rifampicin RF after co-precipitation with different polyethylene glycol (PEG) 6000 ratios. The values are expressed in mean values ± SD, *n* = 3.

**Figure 2 pharmaceutics-13-00336-f002:**
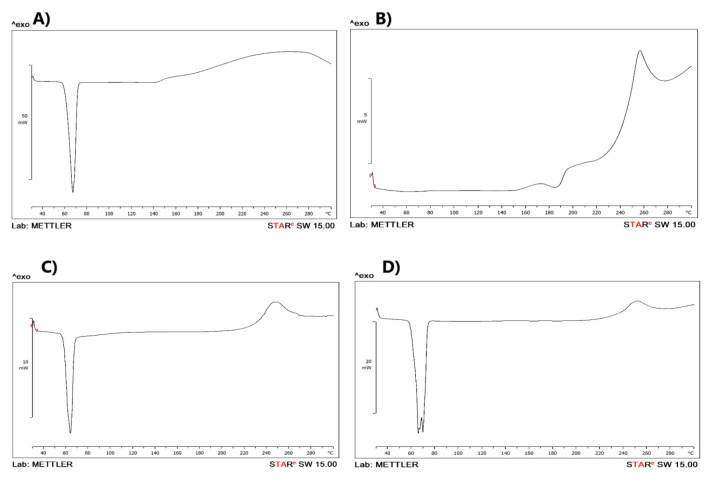
Differential scanning calorimetry (DSC) thermograms of (**A**) PEG 6000, (**B**) RF, (**C**) RF/PEG 6000 co-precipitate and (**D**) RF/PEG 6000 physical mixture.

**Figure 3 pharmaceutics-13-00336-f003:**
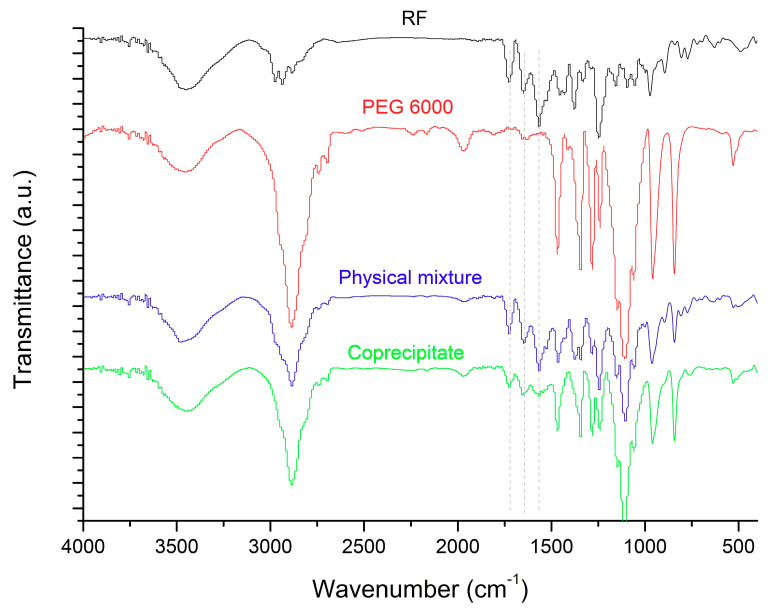
Fourier transform infrared spectroscopy (FTIR) spectra of RF, PEG 6000, the RF/PEG 6000 physical mixture and the RF/PEG 6000 co-precipitate.

**Figure 4 pharmaceutics-13-00336-f004:**
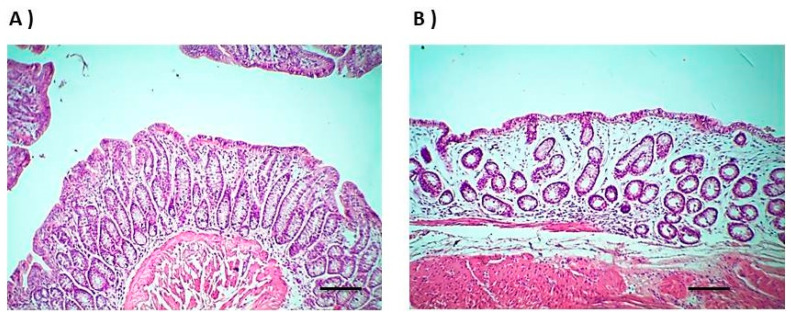
Histological alterations in rectum sections stained by hematoxylin/eosin of; (**A**) normal control rabbits and (**B**) rabbits after the rectal administration of F4 (magnification 100×, bar 100 µm).

**Figure 5 pharmaceutics-13-00336-f005:**
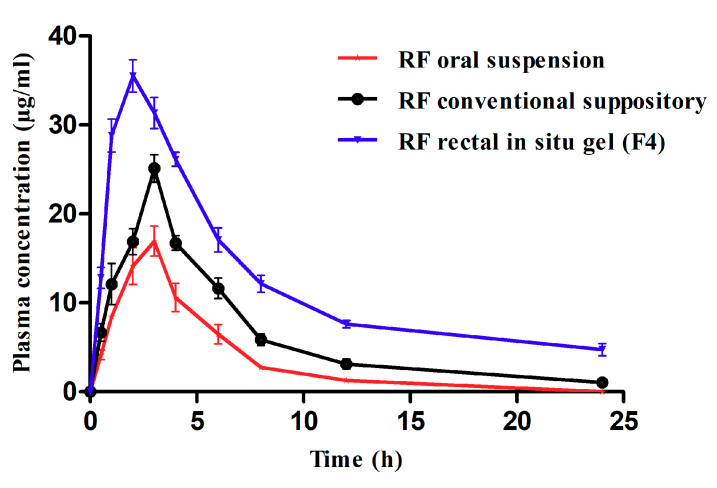
The plasma level-time curve after a single-dose administration of the RF oral suspension, RF rectal conventional suppository and F4. The values are expressed in mean values ± SD, *n* = 6.

**Figure 6 pharmaceutics-13-00336-f006:**
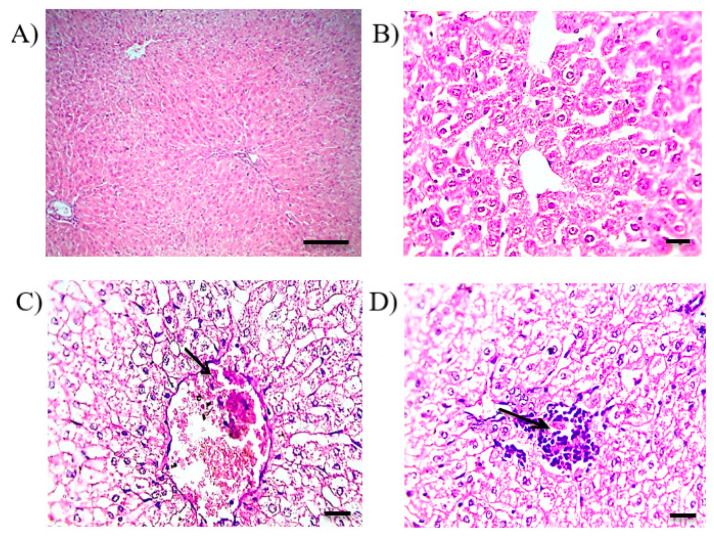
Histopathological alterations in hepatic sections stained by hematoxylin/eosin of; (**A**,**B**) normal control rabbits presenting normal hepatic cells and (**C**,**D**) rabbits that received the RF suspension, showing hydropic degeneration, blood vessel congestion and infiltration by mononuclear inflammatory cells. Magnification for; (**A**) ×100 and bar 100 µm, (**B**–**D**) ×400 and bar 50 µm.

**Figure 7 pharmaceutics-13-00336-f007:**
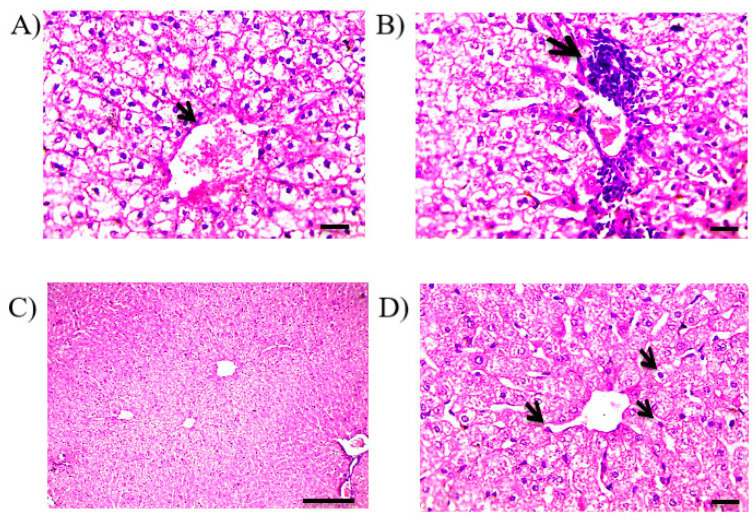
Histological changes in liver sections stained by hematoxylin/eosin of; (**A**,**B**) rabbits that received solid suppositories, showing moderate improvement of hepatic histology and moderate hydropic degeneration and (**C**,**D**) rabbits that received F4, showing mild hepatocytes swelling in centrilobular zone with narrowing of hepatic sinusoids. Magnification for; (**A**,**B**,**D**) ×400 and bar 50 µm, (**C**), ×100 and bar 100 µm.

**Figure 8 pharmaceutics-13-00336-f008:**
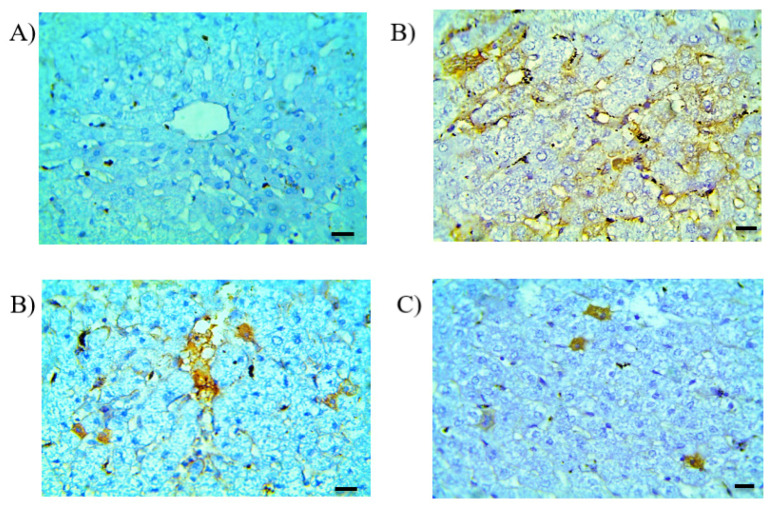
Immunostained liver sections for tumor necrosis factor α (TNF-α) of (**A**) rabbits that received saline presenting an unnoticeable expression of TNF-α, (**B**) rabbits that received the RF oral suspension showing a positive reaction, (**C**) rabbits that received solid suppositories presenting some sporadic cells and (**D**) rabbits administered F4 presenting a minimal positive reaction. Magnification ×400 and bar 50.

**Figure 9 pharmaceutics-13-00336-f009:**
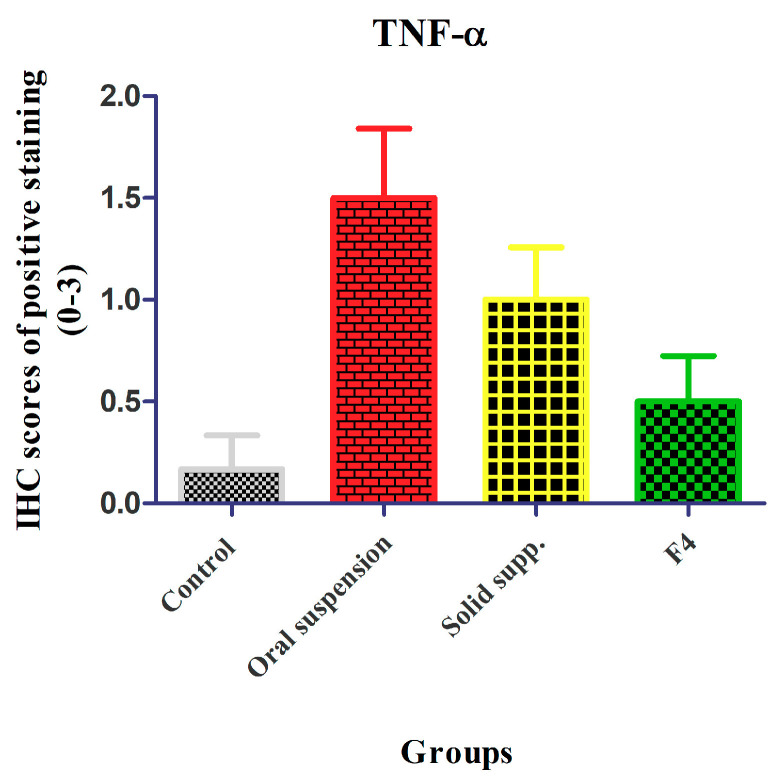
Morphometric analysis of immunohistochemical (IHC) positive staining score of TNF-α reaction in different groups. The values are expressed in mean values ± SD, *n* = 6.

**Table 1 pharmaceutics-13-00336-t001:** Compositions of different in situ rectal gel batches.

Code	Base		Co-Precipitate %	Sodium Alginate %	HPMC %	Chitosan %
	Pluronic F127%	Pluronic F68%				
F1	15	10	0.1	---	---	---
F2	15	10	0.1	0.4	---	---
F3	15	10	0.1	0.8	---	---
F4	15	10	0.1	1.2	---	---
F5	15	10	0.1	---	0.5	---
F6	15	10	0.1	---	1	---
F7	15	10	0.1	---	1.5	---
F8	15	10	0.1	---	---	0.5

**Table 2 pharmaceutics-13-00336-t002:** Gelation temperatures and gel strengths of different in situ rectal gel batches. The values are expressed in mean values ± SD, *n* = 3.

Code	T_sol-gel_ (°C) ± SD	Gel Strength (s) ± SD
Base	36.1 ± 1.5	21.5 ± 1.6	
F1	43.8 ± 3.8	22.4 ± 3.5
F2	40.5 ± 1.9	27.6 ± 0.7
F3	38.5 ± 2.1	34 ± 1.8
F4	34.2 ± 2.7	41.5 ± 2.8
F5	41.3 ± 1.5	29.5 ± 0.9
F6	37.8 ± 1.6	35.9 ± 1.7
F7	Gel formed at room temperature	47.8 ± 3.8
F8	ppt. formed	-

**Table 3 pharmaceutics-13-00336-t003:** Pharmacokinetic parameters of oral RF suspension, rectal RF conventional suppository and in situ rectal gel (mean ± SD, *n* = 6).

Parameter	Oral RF Suspension	Rectal RF Solid Suppository	Rectal RF in Situ Gel (F4)
*T*max (h)	3	3	2
*C*max (µg/mL)	16.9 ± 2.9	25.1 ± 2.7	35.5 ± 3.1
T_1/2ab_ (h)	1.07 ± 0.93	0.15 ± 0.07	4.95 ± 0.35
T_1/2el_ (h)	6.35 ± 2.13	5.20 ± 1.0	7.96 ± 0.8
AUC_0–24_ (µg·h/mL)	86.76 ± 6.8	151.70 ± 12.4	293.76 ± 21.7
MRT (h)	5.38 ± 0.84	6.54 ± 0.27	7.86 ± 1.1
RB	--------	1.74	3.38

**Table 4 pharmaceutics-13-00336-t004:** Serum biochemical parameters in different treated groups (mean ± SD, *n* = 6).

Biochemical Parameter	Control	Oral RF Suspension	Rectal RF Conventional Suppository	Rectal RF in Situ Gel (F4)
ALT (U/L)	48.04 ± 3.8	80.30 ± 0.16	57.30 ± 4.3	50.0 ± 3.9
AST (U/L)	12.71 ± 1.3	37.97 ± 5.9	32.71 ± 4.4	22.26 ± 1.7
Albumin (mg/dL)	4.26 ± 0.35	3.41 ± 0.18	3.80 ± 0.14	3.88 ± 0.16
Total bilirubin (mg/dL)	0.22 ± 0.02	0.61 ± 0.06	0.35 ± 0.05	0.26 ± 0.02

## Data Availability

Not applicable.

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
