# Peer review of "Mucoadhesive In Situ Rectal Gel Loaded with Rifampicin: Strategy to Improve Bioavailability and Alleviate Liver Toxicity"

_pharmaceutics, 2021, doi:10.3390/pharmaceutics13030336_

Round 1

Reviewer 1 Report

Overall, the work is well-written and suitable for publication in Pharmaceutics, However, there are several issues that need revision before publication. 

  • Regarding the methodology, you should try to be more explicit so the work can be reproduce by other authors. For example, how did you do the H&E staining?
  • What is the rational of using PEG6000 to prepare the co-precipitate? HAs any other surfactant be tested?
  • I dont see clearly the explanation about the DSC thermograms. Why did you state that "Shifting the recrystallization exothermic peak into lower temperature confirmed the amorphous attribute of these samples". BUt if they already exhibit a exothermic peak would not be amorphous anyway? Are you referring to a more stable amorphous?
  • In the FTIR figure please indicate with letters which spectra is which. 
  • In figure 5b, I am missing the SD bars.
  • In Table 4 and fig 6 I am missing the statistical analysis. 
  • What do you mean by: "Higher plasma concentrations in case of F4 might be attributed to the dispersability (fluidity) and bioadhesive force variation"?
  • In Table 5, stats are also missing. Have you actually done statistics or not?
  • The discussion is poor, please compare your results with those obtained by other authors. 

Author Response

Comments and Suggestions for Authors

Overall, the work is well-written and suitable for publication in Pharmaceutics, However, there are several issues that need revision before publication. 

Authors’ reply:

The authors would like to thank the reviewer for the comment.

  • Regarding the methodology, you should try to be more explicit so the work can be reproduce by other authors. For example, how did you do the H&E staining?

Authors’ reply:

Regarding section “2.10. Investigation of rectal irritation “, we have added the H&E staining methodology in details. However, in section  “2.13. Toxicity studies” we referred to the procedure in section (2.10) to avoid  the repetition because the method was already mentioned. We have also added the section number between brackets to be more clear. 

  • What is the rational of using PEG6000 to prepare the co-precipitate? HAs any other surfactant be tested?

Authors’ reply:

For solid dispersion preparation, multiple polymers may be applied as:

(a) hydrophilic polymers can be blended with hydrophobic polymers to improve solubility,

(b) hydrophobic polymers can be combined with hydrophilic polymers to improve hydrophobic interactions with poorly soluble drugs, or

(c) a combination of both (a) and (b) strategies may be applied (Eur. J. Pharmaceut. Sci. 106 (2017)

413–421).

PEG 6000 is located in category (a) and has other advantages like improving the molecular interactions with improved stability. The PEG can form hydrogen bonds with both the polymer matrix and the drug molecules (Journal of Controlled Release, 292 (2018) 91-110).

However, we have added this statement in “section  3.1. RF co-precipitate aqueous solubility”.

  • I dont see clearly the explanation about the DSC thermograms. Why did you state that "Shifting the recrystallization exothermic peak into lower temperature confirmed the amorphous attribute of these samples". BUt if they already exhibit a exothermic peak would not be amorphous anyway? Are you referring to a more stable amorphous?

Authors’ reply:

Yes, exactly, we did like to refer to more stable amorphous state of RF. We changed it to be “more stable amorphous state”

  • In the FTIR figure please indicate with letters which spectra is which. 

Authors’ reply:

  1. We have re-drawn the spectra by Origin Software and all spectra were identified at the figure (as seen below).

  • In figure 5b, I am missing the SD bars.

Authors’ reply:

In that figure, we depended upon the averages of the peak areas during the measurements which were collected from UPLC and drawn by external software.

  • In Table 4 and fig 6 I am missing the statistical analysis. 

Authors’ reply:

The values in both were expressed in mean values ± SD, n = 6. Results were analyzed by one-way ANOVA and means were compared by Tukey's multiple comparison testing using GraphPad Prism v.5. Software. Difference at P < 0.05 was considered to be significant. We referred to the significance difference , if any, in the discussion. For example, (F4 provided a significantly (P<0.05) higher AUC0–24 for F4 (293.76±21.7 µg.h/mL) than conventional solid suppository (151.70±12.4 µg.h/mL) and oral suspension (86.76±6.8 µg.h/mL))

  • What do you mean by: "Higher plasma concentrations in case of F4 might be attributed to the dispersability (fluidity) and bioadhesive force variation"?

Authors’ reply:

We have explained both suggestions in the statements came after this sentence and added suitable references. The explanation was as follows: (In case of in situ gel, the dispersion and was more pronounced in the rectal fluids as RF was introduced in soluble co-precipitate form. Bioadhesion and prolonged retention were also higher due to the effect of sodium alginate which improved attachment to rectal mucosa).

  • In Table 5, stats are also missing. Have you actually done statistics or not?

Authors’ reply:

Of course, we have done the statistics using GraphPad Prism v.5. Software. Difference at P < 0.05 was considered to be significant. We referred to the significance difference , if any, in the discussion. For example, (Significant decrease in ALT and total bilirubin was observed in F4 administered group when compared with oral suspension group (p < 0.05)).

  • The discussion is poor, please compare your results with those obtained by other authors. 

Authors’ reply:

We have done more discussion in different sections and added relevant references. All changes were highlighted by yellow color.

Reviewer 2 Report

The current manuscript provides a predictable and no-so-novel account of rectal gel loaded with rifampicin for rectal administration. The rationale and motivation of the research is not clear and the choice of formulation components is questionable. I have following concerns with the study and it design:

  1. Choice of polymers: Alginate, in principle is more of a bioadhesive polymer and that too very mild. The mucoadhesive properties of alginate have always been debatable given the anionic nature of the mucopolysaccharide itself. No mucin interaction studies were provided for the formulation and is a major gap in the study.
  2. Pluronic solutions alone are thermoresponsive only above 20% concentration. The pluronic concentrations in the study are not more than 15% and hence the in situ gelation is debatable. Also, no rheological profiles for temperature ramp studies are provided to support the in situ gelation aspect.
  3. In vivo release studies: The gels are supposed to prolong the release of the drug as the drug molecules will take longer to diffuse through the in situ gel and polymer matrix. The Tmax can be higher but may not be faster than the oral (93% bioavailability has been reported for rifampicin through the oral route) and vice versa. Also it is not clear if the drug alone is going through the membrane or the PEG-drug complex?

Author Response

Comments and Suggestions for Authors

Author reply:

The authors would like to thank the reviewer for the comment.

The current manuscript provides a predictable and no-so-novel account of rectal gel loaded with rifampicin for rectal administration. The rationale and motivation of the research is not clear and the choice of formulation components is questionable.

Authors’ reply:

It is correct that rectal in situ gel has already been utilized for improvement of bioavaiability. However, none of the previous studies in the literature have utilized rectal in situ gel for alleviating rifampicin liver toxicity. Actually, we never mention in the text that combining both effects would be a novel idea. Our aim has from the beginning been to fabricate RF loaded in situ rectal gels in order to minimize acid degradation of the drug following oral administration, improve RF bioavailability and alleviate RF induced hepatotoxicity (lines 78-81). However, it is not at all obvious that using in situ rectal gel would lead to successful results.  Therefore, we screened different the formulation components to select the most appropriate one that can achieve our target.

I have following concerns with the study and it design:

  1. Choice of polymers: Alginate, in principle is more of a bioadhesive polymer and that too very mild. The mucoadhesive properties of alginate have always been debatable given the anionic nature of the mucopolysaccharide itself. No mucin interaction studies were provided for the formulation and is a major gap in the study.

Authors’ reply:

The mucoadhesion process is mediated in two stages. The first stage includes chemical bonding between the mucoadhesive polymer and the mucin. The second stag includes diffusion of the polymer

into the mucus layer (Carvalho et al., Brazilian J. Pharm. Sci. 46 (2010) 1-17). Sodium alginate was reported to interact with mucin the mucus layer by forming hydrogen bonds (Haugstad et al., Polymers, 7 (2015), 161–185). However, for efficient mucoadhesion, the used polymeric chain should be able to penetrate the mucus gel layer close to the epithelium cell lining to form an anchor for the delivery system. This is usually cannot be achieved by a single polymer type and combinations of different polymers with in situ gelling and mucoadhesive properties are needed to attain the desired duration of mucoadhesion (Zahir et al., Exp. Op. Drug Del. 15 (2018) 1007–1019). Additionally, this combination will be more practical to avoid possible liquefaction of the gel layer by the liquid turnover of the gastric mucosa. In our case we combined the effect of in situ gelling polymer (Pluronic) and mucoadhesive polymer (alginate) to improve the mucous layer interaction.

  1. Pluronic solutions alone are thermoresponsive only above 20% concentration. The pluronic concentrations in the study are not more than 15% and hence the in situ gelation is debatable. Also, no rheological profiles for temperature ramp studies are provided to support the in situ gelation aspect.

Authors’ reply:

We used Pluronic in 25% as total concentration. As we can see in table 1, we fabricated different batches with 15% Pluronic F127 and 10% Pluronic F68 (totally 25%).

  1. In vivo release studies: The gels are supposed to prolong the release of the drug as the drug molecules will take longer to diffuse through the in situ gel and polymer matrix. The Tmax can be higher but may not be faster than the oral (93% bioavailability has been reported for rifampicin through the oral route) and vice versa. Also it is not clear if the drug alone is going through the membrane or the PEG-drug complex?

Authors’ reply:

Of course slow diffusion of drug molecule is one of the factors that can control drug release and hence its absorption. Particularly for poorly water soluble drugs, incorporation into fatty based suppositories could slow down their release into rectal fluid because such drugs prefer base matrix over the aqueous rectal fluid. On the other hand, incorporation of RF in more soluble form (coprecipitate) can enhance drug release. Moreover, PEG, as efficient excipient, can improve drug absorption (Li et al., Int. J. Pharm. 403 (2011) 37-45).So, it is suggested that combination of all these factors could be the reason for fast absorption of in situ gel formulation.

We have explained this suggestion in PK section (3.9. Pharmacokinetic studies).

However, PEG 6000 can improve drug absorption but it itself cannot be absorbed because its mol weight is above 1000.      

Reviewer 3 Report

General

The article is scientifically interesting but needs to be improved to be published. One of the aspects has to do with the discussion of the results in which it is verified that there is a need for greater reference to works already in the area.

There is a part that is not justified, which is the validation of the chromatographic method of RF quantification, reference to the linearity range, the limit of quantification and detection are sufficient thus, the entire section 3.8. Chromatography should be deleted from the article.

Specific points:

2.1. Materials

The characteristics presented are the specifications, what are the characteristics of the chitosan used in the experiments (molecular weight/viscosity ). This data can be obtained using the batch number from sigma Chitosan samples.

The characteristic of sodium alginate (SA) must be added, namely MW/viscosity/% of different monomers G and M α-l-guluronic acid (G) and β-d-mannuronic acid (M) residue.

Line 149

….weighing thirty five gm…

Correct the gm for grams or if use the abbreviation it must comply with SI (g).

Line 163

“Rectal irritation of the selected formulation was assured depending upon histopathological examination of rectal specimens after administration of the formulation.”

What you mean? That the rectal irritation of the selected formulation was assessed? Please rephrase accordingly.

3.2. Thermal analysis

The Figure 2. DSC thermograms needs to be repeated the assay for the (B) RF, the peack of RF exhibited a melting endotherm in the range of 180 °C to 200 °C (186.3 °C) is not visible, probably low amount of sample was used.

With the results presented is not possible to conclude the state in terms of crystal state of the RF from the figure 2 results.

3.8. Chromatography

This can be removed from the paper and resumed to the reference to the linearity range, the limit of quantification and detection.

Figure 3 needs to be improved the assignment of the spectra to the different samples is difficult of reading.

Figure 4 is missing the scale bar on the images.

On the legend of Figures 7 and 8 is missing the units of the scale bar?

Author Response

Comments and Suggestions for Authors

General

The authors would like to thank the reviewer for the comment.

The article is scientifically interesting but needs to be improved to be published. One of the aspects has to do with the discussion of the results in which it is verified that there is a need for greater reference to works already in the area.

Authors’ reply

We have done more discussion in different sections and added relevant references. All changes were highlighted by yellow color.

There is a part that is not justified, which is the validation of the chromatographic method of RF quantification, reference to the linearity range, the limit of quantification and detection are sufficient thus, the entire section 3.8. Chromatography should be deleted from the article.

Authors’ reply

Ok, It has also recommended by reviewer 1 so, we have removed this part from the manuscript and it has been added as supplementary data.

Specific points:

2.1. Materials

The characteristics presented are the specifications, what are the characteristics of the chitosan used in the experiments (molecular weight/viscosity ). This data can be obtained using the batch number from sigma Chitosan samples.

Authors’ reply

The molecular weight of chitosan is 100,000-300,000 Da and degree of deacetylationis 85%. This information is already present in the manuscript. We have added the viscosity according to the manufacturer (viscosity: 200-800 cP ).

The characteristic of sodium alginate (SA) must be added, namely MW/viscosity/% of different monomers G and M α-l-guluronic acid (G) and β-d-mannuronic acid (M) residue.

Authors’ reply:

The specifications were added as follows: molecular weight: 80,000-120,000 Da with mannuronic/guluronic ratio of about 1.56; viscosity: 3.500 cP. 

Line 149

….weighing thirty five gm…

Correct the gm for grams or if use the abbreviation it must comply with SI (g).

Authors’ reply:

Ok. We have corrected.

Line 163

“Rectal irritation of the selected formulation was assured depending upon histopathological examination of rectal specimens after administration of the formulation.” 

What you mean? That the rectal irritation of the selected formulation was assessed? Please rephrase accordingly.

 Authors’ reply:

Ok. We have rephrased the statement to be:

“Safety of rectal administration of the selected formulation was assured by examination of rectal specimens and detection of histopathological changes (rectal irritation)”.

3.2. Thermal analysis

The Figure 2. DSC thermograms needs to be repeated the assay for the (B) RF, the peack of RF exhibited a melting endotherm in the range of 180 °C to 200 °C (186.3 °C) is not visible, probably low amount of sample was used.

With the results presented is not possible to conclude the state in terms of crystal state of the RF from the figure 2 results.

 Authors’ reply:

The endothermic peak is not predominant when we made overlay of different curves in one panel by the software (STARe SW) because Y-axis was adjusted by the software to figure out all peaks with different heat flow. However, the peak is more clear in single RF curve. We have marked the peak by the arrow.

The image of RF from the software seems to be better (image below) when presented as single image.

However, we have merged single images into one panel to be more relevant to our explanation (image below).

On the other hand, we did like to refer to more stable amorphous state of RF. We changed it to be “more stable amorphous state”

3.8. Chromatography

This can be removed from the paper and resumed to the reference to the linearity range, the limit of quantification and detection.

 Authors’ reply:

Ok, and as recommended by Reviewer 1, we have removed this part from the manuscript and it has been added as supplementary data.

Figure 3 needs to be improved the assignment of the spectra to the different samples is difficult of reading.

 Authors’ reply:

  1. We have re-drawn the spectra by Origin Software and all spectra were identified at the figure (as seen below).

Figure 4 is missing the scale bar on the images.

 Authors’ reply:

Ok. We have added the bar on the image and changed the legend to include the data (magnification 100x, bar 100 µm).

On the legend of Figures 7 and 8 is missing the units of the scale bar?

Authors’ reply:

Ok. We have added the scale bar to the legend to include the data.

Round 2

Reviewer 2 Report

The authors tried to provide rebuttals for the comments raised by the reviewer and I agree with most of the responses. The authors still need to address two major gaps in the study:

  1. No mucin interaction studies were provided for the formulation and is a major gap in the study.
  2. No rheological profiles for temperature ramp studies are provided to support the in situ gelation aspect.

Author Response

The authors tried to provide rebuttals for the comments raised by the reviewer and I agree with most of the responses. The authors still need to address two major gaps in the study:

  1. No mucin interaction studies were provided for the formulation and is a major gap in the study.

Autors’ reply:

Actually we don’t have to study the mucoadhesion power of the prepared formulation as we have already added different mucoadhesive polymers. All added mucoadhesive polymers were already examined for mucoadhesion and well documented in that regard. So, no need to repeat such already done work. Please check the publications below:

  • International Journal of Pharmaceutics 430 (2012) 114– 119
  • Pharm. Res. (2013) 36:586–592
  • Eur J Drug Metab Pharmacokinet (2014) 39:283–291
  • AAPS PharmSciTech 2006; 7 (2) Article 38
  • APPS PharmSciTech, Vol. 10, No. 3, September 2009
  • AAPS PharmSciTech (# 2017), DOI: 10.1208/s12249-017-0839-5

  1. No rheological profiles for temperature ramp studies are provided to support the in situ gelation aspect.

Autors’ reply:

As we stated in “Introduction”-section, the study aimed to fabricate RF loaded in situ rectal gels in order to minimize acid degradation of the drug following oral administration, improve RF bioavailability and alleviate RF induced hepatotoxicity. So, gelation temperature and gel strength are the most important physicochemical properties that assures the conversion of the formulation into gel upon insertion into rectum (thermogelation) and indicates easiness of insertion/rectal escape of the suppositories respectively. Therefore, we see that study of rheological profiles will be of little usefulness based on the aim of the work.  

Reviewer 3 Report

The authors answered all questions. The article has been improved by what can be considered for publication. 

Author Response

The authors would like to thank the reviewer.

Round 3

Reviewer 2 Report

No further comments. It appears that the authors are not interested in carrying out the additional tests.

If the mucoadhesiveness is know for the formulation; then the rationale of this work and hence novelty is questionable. This makes the work predictable.

The rheological data via temperature ramping would have added a lot of value. The magnetic stir protocol is subjective. One rheo curve can provide both the exact gelation temperature as well the gel strength over the temperature range. This will be in line with what authors said "So, gelation temperature and gel strength are the most important physicochemical properties that assures the conversion of the formulation into gel upon insertion into rectum (thermogelation)". Methods and analyses stating that "Gelation temperature is considered when the magnetic bar stopped moving" and "The time (s) passed to drop the disk by five cm down was recorded and considered as an arbitrary index of gel strength." cannot provide accurate data and as such are arbitrary.

It is worth noting that these two aspects are mentioned in the title of the manuscript and hence are important.

This manuscript is a resubmission of an earlier submission. The following is a list of the peer review reports and author responses from that submission.

Round 1

Reviewer 1 Report

The research idea lacks for novelty. There is a large similarity between the present work and that had been published in Japanese Journal of Pharmaceutical Health Care and Sciences. (2004) 30 (9):574-583 (DOI: 10.5649/jjphcs.30.574)

Reviewer 2 Report

The paper deals with the development and characterization of an in situ gelling formulation for the rectal delivery of rifampicin.  The performance of the developed formulation in terms of bioavailability and toxicity was evaluated in an in vivo animal model.

Even if many articles has been already published on the development of in situ gelling formulations based on pluronics, the in vivo animal studies performed on the developed formulation makes the topic interesting.

The paper has to be revised before pubblication. In particular, the in vitro characterization of the developed formulation and the presentation of the relevant results should be improved.

Specific comments and requests of revision are hereafter reported.

Introduction

Lines 60-66: The authors should rewrite the paragraph. As it is written, it is not clear. Moreover. strategies on the employment of poloxamers in rectal in situ gelling formulations should be described and relevant references should be cited.

Line 68: “….. to fabricate RF in  thermoresponsive mucoadhesive liquid suppositories…….” the sentence has to be rewritten.

Materials and methods

Lines 77-82: Information about the chitosan grade (source, MW) has to be added. Was chitosan base or a salt used?

Line 96: “Based on aqueous solubility results, 1:1 of RF: PEG 6000 ratio was selected for further investigation.” This is a result. Such a sentence should be removed from Materials and methods section.

Line 111- 118: Was chitosan soluble in the vehicle?

Line 120- 127: A rheological analysis should be performed to better investigate the formulation capability to gelify as a response of temperature increase. Such analysis could be profitably employed also to evaluate the gel strength.

Lines 140-145: “Based on results of aforementioned experiments, F4 was selected for all in vivo experiments. 144 Four albino rabbits were rectally administered…” This is a result. Such a sentence should be removed from Materials and methods section.

Results and discussion

Line 241: “It is also obvious that RF solubility enhanced by co-precipitation with PEG” This sentences has to be rewritten.

Figure 1: What is the ratio between RF and PEG in the physical mixture? Different physical mixtures prepared according to the various RF-PEG ratios employed in the co-precipitates should be tested. Information about the number of replicates has to be added (mean values ± …., n= …). Do the error bars refer to s.e or s.d.?

Line 250: “RF exhibited a melting endotherm in the range of 180 °C to 200 °C 250 (186.3 °C)”. The endothermal peak is not evident.

Line 291-2: The mechanism of Pluronic gelation due to micelles packing has to be better explained.

Table 2 – The Unit of the gelation temperature has to be added (°C).  Information about the number of replicates has to be added.

Line 425: “Absorption via lower hemorroidal vein bypassed hepatic first pass effect.” It is not completely true due to vein anastomosis. The sentence has to be rewritten.

Figure 6 - Information about the number of replicates has to be added in the Figure legend (mean values ± …., n= …).

Figure 10 - Information about the number of replicates and the identification of the error bars (s.e. or s.d.) have to be added in the Figure  legend (mean values ± …., n= …).

Reviewer 3 Report

The manuscript submitted by Fakhria Al-Joufi, Mohammed Elmowafy, Nabil K. Alruwaili, Khalid S. Alharbi, Khaled Shalaby, Shaker Alshararee, Hazim M. Ali described the „Mucoadhesive in situ gelling rifampicin suppositories; strategy to improve bioavailability and alleviate liver toxicity” presents a new and interesting subject.

Reviewer’s opinion/remarks/question regarding the revised manuscript:

  • Instead of the oral administration of rifampicin, rectal administration may indeed be beneficial in terms of both absorption efficiency and hepatotoxicity.
  • Unfortunately, as regards the name of the dosage form, the authors themselves use different nomenclature: in situ gelling rifampicin suppositories, liquid suppositories were developed, in situ rectal gel, the latter being closest to reality, as the liquid introduced into the rectum will gel, so I would not use the term suppository either in the title or in the article.
  • In the “Introduction” (line 43), the authors mention the active ingredient they want to use and immediately give its abbreviation, which is all right /Rifampicin (RF)/, but in Section “2.1. Materials” the full name of the active ingredient should be given again (line 78).
  • Also in line 78, the origin of the active ingredient is given as “CDH”, which does not clearly refer to the manufacturer. It is essential to specify the abbreviation “CDH”.
  • In Section “Development of RF co-precipitate” (line 83), an organic solvent (chloroform) was used to prepare the co-precipitate, but it was not verified whether there was any of it left in the sample or not (lines 84-89).
  • In Section 2.4.1., in lines 101-102, it is written: “Accurately weighed 6 milligrams of each sample were put in crucible aluminum pans”, which makes the work unnecessarily difficult, as for better comparability we use mass normalization for plotting DSC curves (W/g).
  • In Section “2.5 Preparation of in situ gels”, in lines 117-118, it is written correctly that Table 1 shows the composition of the different samples, but the title of the table is incorrect because “Solubility of RF after coprecipitation with different PEG 6000 ratios” is written, but this is the title of Figure 1.
  • Table 1 gives the percentage of solid components, is the rest all water then?
  • From Section “2.5 Preparation of in situ gels” we know that the concentration of RF is 50 mg/ml, but it is not given how many ml were used at a time.
  • In Section “3.2. Thermal analysis” the authors write that: “RF exhibited a melting endotherm in the range of 180 °C to 200 °C (186.3 °C), directly followed by an exothermal recrystallization at 256.2 °C.” (lines 250-251), but the endothermic phenomenon is hardly visible, let alone the melting point, and the exothermic phenomenon is not immediately after the endothermic one, but at a temperature 50-60 °C higher, which is already an exothermic signal of decomposition. The authors try to support what is described in lines 250-254 with reference 21, but there a heating rate of 10 °C/min was used instead of 20 °C/min. The change in the heating rate also changes the shape of the peaks, and there is a difference particularly in the appearance of the phenomena because at a higher rate the peaks shift towards a higher temperature.
  • Thermoanalytics is not always sufficient to prove the amorphous/crystalline form, especially without thermogravimetric (TG) measurements, the XRPD device is much more suitable.
  • I do not know how important the crystalline form of the active ingredient is, as it will be used in a liquid medium.
  • How do the authors explain the double endothermic peak seen in curve “D” in Figure 2? (RF/PEG 6000 physical mixture). At this temperature the active ingredient has no peak and PEG 6000 should have one peak.